# Mineral Interpretation Discrepancies Identified between Infrared Reflectance Spectra and X-ray Diffractograms

**DOI:** 10.3390/s21206924

**Published:** 2021-10-19

**Authors:** Fardad Maghsoudi Moud, Fiorenza Deon, Mark van der Meijde, Frank van Ruitenbeek, Rob Hewson

**Affiliations:** Department of Earth Systems Analysis (ESA), Faculty of Geo-Information Science and Earth Observation (ITC), University of Twente, 7514 AE Enschede, The Netherlands; f.deon@utwente.nl (F.D.); m.vandermeijde@utwente.nl (M.v.d.M.); f.j.a.vanruitenbeek@utwente.nl (F.v.R.); Hewson001@gmail.com (R.H.)

**Keywords:** shortwave infrared, X-ray powder diffraction, discrepancy, clay mineral interpretation, hydrothermal alteration minerals

## Abstract

Mineral composition can be determined using different methods such as reflectance spectroscopy and X-ray diffraction (XRD). However, in some cases, the composition of mineral maps obtained from reflectance spectroscopy with XRD shows inconsistencies in the mineral composition interpretation and the estimation of (semi-)quantitative mineral abundances. We show why these discrepancies exist and how should they be interpreted. Part of the explanation is related to the sample choice and preparation; another part is related to the fact that clay minerals are active in the short-wave infrared, whereas other elements in the composition are not. Together, this might lead to distinctly different interpretations for the same material, depending on the methods used. The main conclusion is that both methods can be useful, but care should be given to the limitations of the interpretation process. For infrared reflectance spectroscopy, the lack of an actual threshold value for the H–OH absorption feature at 1900 nm and the poorly defined Al–OH absorption feature at 2443 nm, as well as for XRD, detection limit, powder homogenizing, and the small amount of montmorillonite below 1 wt.%, was the source of discrepancies.

## 1. Introduction

Mineral composition interpretation with methods such as reflectance spectroscopy and X-ray diffraction (XRD) can potentially identify hydrothermal alteration zones [1,2,3]. From laboratory to spaceborne scale, infrared reflectance spectroscopy has been used to identify different clay minerals within the hydrothermal alteration zones (e.g., [1,2,3,4]). Moreover, XRD has been used in several studies to identify clay minerals within the different soil and rock types (e.g., [5,6,7]). Shortwave infrared (SWIR) reflectance spectroscopy, a nondestructive technique, provides rapid results by measuring spectral absorption features caused by molecular bonding mechanisms within a depth of the first few nanometers below the sample surface. XRD detects the lattice spacing of a crystalline structure of the minerals, penetrating the interior of the mineral structure. XRD has been applied for reflectance spectroscopic validation [1,2,3]. In particular, XRD has been used to validate identified hydrothermal alteration clay minerals with SWIR reflectance spectroscopy [1,2,3,4]. However, in some cases, comparing mineral maps obtained from reflectance spectroscopy with XRD has shown inconsistencies in the mineral composition interpretation and the (semi-)quantitative mineral abundances [1,2,3,4]. The reasons for observing these disagreements have been insufficiently investigated. This study aims to answer the following research questions: (a) Why do these discrepancies exist? (b) How should they be interpreted? A careful selection of two out of 19 rock samples from different altered lithological units within the Kuh Panj porphyry copper occurrence located within the southeastern part of Iran (for further information, refer to [3]) were investigated to provide answers to these questions. The samples were collected from different hydrothermal alteration zones from argillic to phyllic but within the same lithological unit of andesite and dacite [3]. Since all samples were taken from the andesite and dacite unit, selecting the different hydrothermal alteration zones was considered as the sample selection criterion.

## 2. Materials and Methods

This section outlines the details of each dataset utilized in this study, considered important for evaluating the findings.

### 2.1. Preparation of Rock and Powder Samples

Rock samples were gently crushed with a hammer, ground into powders using a pestle and mortar, and sieved to −80 mesh powders. The powders were prepared for two types of measurements: (a) XRD patterns from the whole rock to identify the dominant minerals, and (b) low-angle patterns from the extracted clay fraction to determine the clay mineral types.

The powdered rock samples were placed in a 1 L cylinder with 1000 cc of deionized water and 10 cc of sodium hexametaphosphate in a bottle. The sodium hexametaphosphate acted as a dispersive agent and facilitated the dispersion of the small particles in the solution [5,6]. The solution was centrifuged for 16 h to ensure that all particles were separated. Then, the clay particles (≤2 µm) were extracted following the pipette method [7]. In this method, which is a gravitational method, particles were separated on the basis of their density. After 6 h sedimentation time, particles with ≤2 µm particle size were extracted using a pipette in the top 200 cc. The separated fraction was dried in an oven for 24 h at 25 °C. The sedimentation of the previous stage was dissolved with 20 mL of deionized water and placed in a 2 µm filter on a vacuum pump. The solution was pumped to rinse the sodium hexametaphosphate from the clay particles. The clay fraction was dried at 25 °C for 24 h. The dried clay fraction was measured with low-angle XRD measurement.

### 2.2. SPECIM Data Acquisition and Mineral Mapping

The SPECIM camera (https://www.specim.fi/products/swir/ (accessed on 18 October 2021)) with an OLES30 lens, 30 cm measurement distance from the rock samples, 256 µm pixel size, and 98 mm image swath was used to acquire hyperspectral images. The SPECIM camera captures SWIR hyperspectral images from 940 nm to 2540 nm with 6 nm spectral resolution and a 0.2 mm spatial resolution. SPECIM hyperspectral images were acquired from both the weathered surface and a freshly cut surface of each sample to compare the mineralogical composition of both surfaces.

Spectral endmembers (compositional components defined by diagnostic spectral features) were collected using the wavelength mapping method [8]. The endmember interpretation strategy was described by Maghsoudi et al. [3]. A decision tree was developed to classify the SPECIM images into mineral maps (Figure 1) [9]. The classifier uses the wavelength position of absorption features and their absorption depth [10,11,12]. The decision tree was used because it helps to include and evaluate the depth of specific absorption features in the classification procedure. Minerals without absorption features in the SWIR, such as quartz, could not be detected and mapped with the SPECIM images. However, any pixel containing a small proportion of SWIR-active minerals, such as clay minerals, shows absorption features and is mapped by SPECIM hyperspectral images even though those pixels also contain non-SWIR-active minerals such as quartz [13,14]. The number of pixels in a class as a function of the total rock pixels was used to compute the abundance of each spectrally active mineral.

All Al-bearing phyllosilicate minerals have an Al–OH absorption feature in the range of 2100 to 2300 nm. Hence, in the first stage of the decision tree, they were separated from non-Al–OH-containing minerals and the background. A <3% absorption feature was defined as the threshold limit for noise with >3% defined as an Al–OH absorption feature. We selected 3% absorption as noise by computing the signal-to-noise ratio and checking noise variation within the whole SWIR spectral range, particularly wavelengths with no absorption feature within the weathered surface and freshly cut surface of the rock samples. Therefore, we considered any feature with more than 3% absorption as an absorption feature. Montmorillonite has no absorption feature from 2300 to 2500 nm and has a deeper H–OH absorption feature at 1900 nm than the Al–OH absorption feature near 2200 nm [10]. However, montmorillonite + illite and illite both have an Al–OH absorption feature in common at approximately 2343 nm [15]. Depending on the relative proportion of illite in the montmorillonite + illite mixture, montmorillonite + illite could have a deeper, shallower, or sometimes lack of an Al–OH absorption feature at approximately 2443 nm [1,2]. If a pixel spectrum lacked an Al–OH absorption feature from 2400 to 2500 nm, that pixel was classified as montmorillonite + illite. The crystallinity spectral ratio was used to discriminate montmorillonite + illite with a shallow Al–OH absorption feature from illite. Montmorillonite + illite has a deeper H–OH absorption feature at 1900 nm than the Al–OH absorption feature of illite, approximately 2209 nm. By dividing the depth of the Al–OH absorption feature over the depth of the H–OH absorption feature, the crystallinity degree can be determined. If this ratio is equal to or higher than one, it means that the pixel spectrum has an equal or shallower H–OH absorption feature than the Al–OH absorption feature, which classifies it as illite. On the other hand, if this ratio is lower than one, the pixel spectrum has a deeper H–OH absorption feature than the Al–OH absorption feature (e.g., the montmorillonite spectrum mentioned earlier in this paragraph), which classifies it as montmorillonite + illite. A lower spectral crystallinity ratio denotes a relatively greater proportion of montmorillonite in the montmorillonite + illite spectrum.

### 2.3. XRD Setup

An Ni-filtered Cu-Kα radiation tube with 1.54184 [Å] and a 2θ scanning of 0.01/s for 30 rounds was set to 30 kV voltage and 10 mA current for XRD measurements via a Bruker D2 Phaser (https://www.bruker.com/products/x-ray-diffraction-and-elemental-analysis/x-ray-diffraction/d2-phaser.html (accessed on 18 October 2021)). The scanning step was from 6° to 80° (2θ) for the whole-rock XRD patterns and from 5° to 25° (2θ) for identifying the minerals forming the clay fractions. Clay minerals often show the most intense and sharp reflection in the range of 5° to 25° (2θ). Two discriminators with the lower level at 0.18 mm and the upper level at 0.25 mm were used. A divergent slit of 0.06 mm, a detector slit of 8 mm, and a 1 mm knife were used in order to minimize scattering effects. Diffractograms were interpreted semi-quantitatively with the DIFFRAC.EVA software version 6 of Bruker D2 Phaser and a library to assign the respective minerals to the peaks [16].

## 3. Results

### 3.1. SPECIM Mineral Maps

Montmorillonite, illite, and their mixture were the SWIR-active mineral compositions of the rock samples. The classified SPECIM images showed that SWIR-active minerals observed on the weathered surface also existed on the freshly cut surface (Figure 2), except for montmorillonite. This suggests that montmorillonite formed as a surface coating due to weathering processes (Figure 2) [17,18,19]. Illite and a montmorillonite + illite mix (interpreted on the basis of individual SPECIM pixels) were the dominant mineral classes observed in the samples. A comparison of the illite spectra of both samples demonstrated that the illite-dominant samples (Figure 2a) had a relatively higher spectrally determined illite crystallinity than the montmorillonite + illite-dominant samples (Figure 2b). In contrast to montmorillonite, which lacks absorption features between 2300 and 2500 nm, and illite, with two absorption features at approximately 2345 and 2435 nm, the montmorillonite + illite mixture had only one absorption feature at 2345 nm [1,2,3].

### 3.2. XRD Mineral Composition

The whole-rock XRD patterns showed albite, quartz, and illite/muscovite (Figure 3a,b). Albite had three prominent peaks at approximately 14°, 22°, and 27° 2θ in both samples. Moreover, quartz was identified by an intense, sharp peak in both samples at the 26° 2θ peak.

The patterns collected on extracted clay fractions evidenced the presence of illite/muscovite (Figure 3). Since muscovite was not observed within the reflectance spectra, the peak at 8°–9° 2θ was matched with illite. Although montmorillonite and montmorillonite + illite were observed with reflectance spectroscopy, the expected montmorillonite peak at approximately 6°–7° 2θ [16] was not in the patterns.

## 4. Discussion

In this study, SPECIM and XRD results presented inconsistencies in the mineral interpretation and their (relative) abundances. Previous studies (e.g., [1,2,3,4]) also reported a conflict between XRD and reflectance spectroscopic results. However, the reasons for these discrepancies were not investigated. In our study area, Ayoobi and Tangestani [21] used XRD and detected montmorillonite in the area.

From a mineral structure perspective, montmorillonite and montmorillonite + illite contain more interlayered H–OH content in their structure than illite [11,12]. This difference is observable via infrared reflectance spectroscopy by showing a deeper absorption feature at approximately 1900 nm for montmorillonite and montmorillonite + illite compared to the illite and an Al–OH absorption feature at approximately 2209 nm. However, due to gradual changes in the H–OH absorption feature from montmorillonite + illite to illite [3], there is no actual threshold for discrimination of these two minerals on the basis of the depth of the H–OH absorption feature. A poorly defined Al–OH absorption feature at 2443 nm for illite due to differences in the polyhedral geometrics of illite leads to misinterpretation of illite with montmorillonite + illite [12]. However, as XRD peaks correspond to a specific lattice plane within the mineral structure in which montmorillonite (d-spacing: 15 Å) and illite (d-spacing: 10 Å) have different lattice spacing [16], they can be easily distinguished, even when occurring as a mixture.

X-ray beams and infrared reflectance spectroscopy techniques are capable of sensing to the depth of approximately 2 and 0.0012 mm (half of the used wavelength, in our case, with a maximum of 2400 nm considered) of particles with an approximate size of 63 microns [22,23]. In this study, montmorillonite showed a 6% SWIR-active weathered surface mineral proportion, but only 0.1% in the freshly cut surface. The montmorillonite surface coverage area was approximately equal to 0.16 wt.% (see the equation below) of the whole rock, which is lower than the 1–4 wt.% detection limit of XRD [24,25] to be detected in the powders.

Montmorillonite wt.% = 0.256 mm SPECIM pixel length × 0.256 mm SPECIM pixel width × 900 number of montmorillonite pixels in weathered surface sample × 0.0012 mm spectroscopic penetration × 2.35 kg/m^3^ montmorillonite density.

Since spectrally interpreted montmorillonite was merely present as a thin surface coating due to weathering of the rock sample [19,26], the homogenized nature of the sample powder and collection for the XRD measurement may have resulted in a concentration below the detection limit of 1 wt.%, making montmorillonite difficult to detect. The crystallinity properties of the clay minerals might have been affected by the drying (period) and powdering procedure, leading to a change in the crystalline structure of clay minerals [26,27,28,29], resulting in poorly or absent defined peaks in the diffractograms and, thus, hampering the detection of montmorillonite by XRD.

This could be another reason why the semi-quantitative abundances estimated with XRD do not match with the SPECIM result. The whole-rock XRD results showed an average of 28% quartz, 22% albite, and 50% sheet silicate proportions within the rock powders. In contrast, SPECIM mineral maps showed that more than 70% surface area of the samples contained clay minerals (montmorillonite, illite, and mixture). This inconsistency occurred because (a) any pixel with a small proportion of an SWIR-active mineral would have an SWIR-active reflectance spectrum in SPECIM, and (b) with the SPECIM camera used, it is not possible to identify and map non-SWIR-active minerals (e.g., thermal infrared diagnostic quartz and feldspars). Therefore, the SPECIM-derived mineral quantifications only cover SWIR-active minerals, while XRD mineral abundances refer to both SWIR- and non-SWIR-active minerals. The semi-quantitative results disagreed because non-SWIR-active minerals (i.e., quartz) were not mapped with reflectance spectroscopy.

## 5. Conclusions

This study discussed the different reasons for observing discrepancies in the results of the mineral composition and semi-quantitative interpretation between X-ray diffraction (XRD) and infrared reflectance spectroscopy. The absence of an actual reflectance value threshold for the H–OH absorption feature at approximately 1900 nm and a poorly defined Al–OH absorption feature at approximately 2443 nm lead to an uncertain interpretation of montmorillonite + illite and illite with infrared reflectance spectroscopy and an discrepancy with the XRD output. X-ray and infrared reflectance spectroscopy penetrate to a 2 and 0.0012 mm depth of particles, respectively, with an approximate size of 63 microns. XRD has a detection limit of about 1–4 wt.% and could not detect the estimated 0.16 wt.% coating of montmorillonite detected with SPECIM, which was formed on the surface of a rock sample due to weathering. The semi-quantitative mineral composition inconsistency between XRD and infrared reflectance spectroscopy exists because the SPECIM platform only covers the shortwave infrared (SWIR; due to the focus of this work on clay minerals) range, in which clay minerals are detectable and non-SWIR-active minerals such as quartz and albite are impossible to be mapped, detected, and quantified.

Using the XRD-measured powder for reflectance spectroscopy can mitigate the discrepancies described in this study. Ethylene glycol treatment is also suggested to be applied on samples for montmorillonite/chlorite identification. This research can benefit geoscientists such as soil scientists, mineralogists, and geologists who work on mineral exploration projects by improving the confidence of mapping hydrothermal alteration zones defined by the presence or absence of montmorillonite and illite.

## Figures and Tables

**Figure 1 sensors-21-06924-f001:**
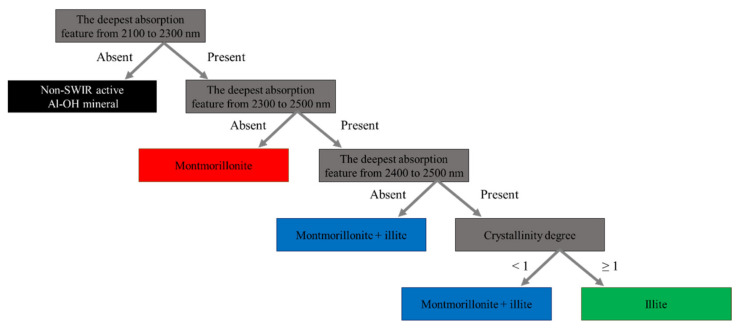
The decision tree for SWIR-active mineral classification. The decision tree was constructed on the basis of the spectral properties of identified minerals within the weathered and freshly cut surface of the rock samples. Montmorillonite + illite indicates a mixture of montmorillonite and illite [1,2,3].

**Figure 2 sensors-21-06924-f002:**
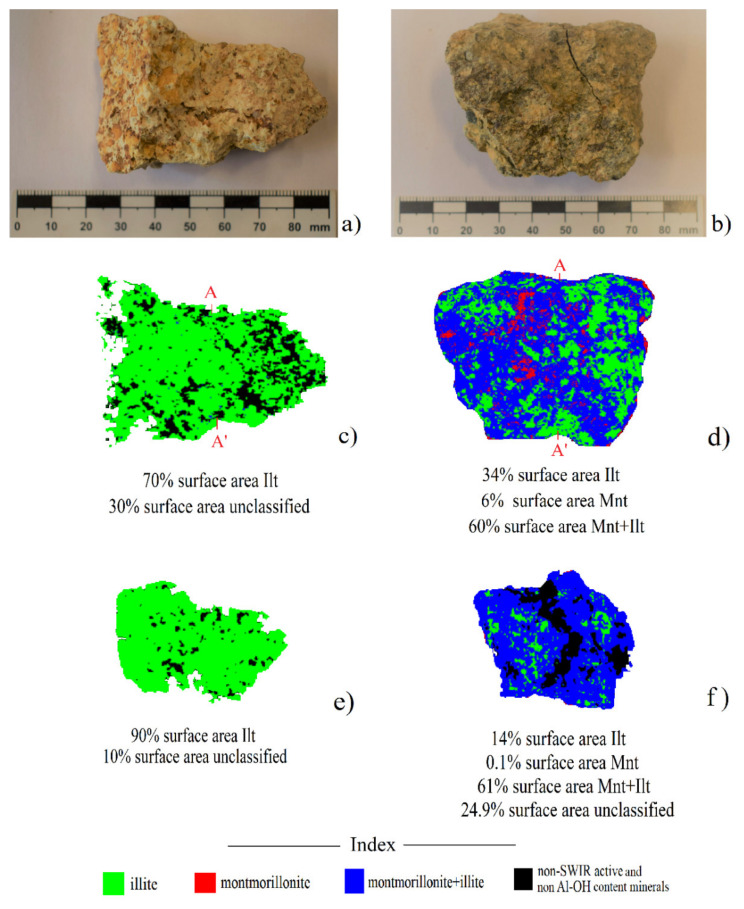
Natural color of samples S08 (**a**) and S06 (**b**). Mineral maps and semi-quantitative mineral abundances in surface area percentage for the weathered surface of S08 (**c**). A/A′ shows the cutting location of the rock sample and S06 (**d**), and the freshly cut surface of S08 (**e**) and S06 (**f**). Average hull removal SWIR spectra of rock samples in both weathered and freshly cut surfaces (**g**). Abbreviations based on Whitney and Evans [20]. Ilt = illite; Mnt = montmorillonite.

**Figure 3 sensors-21-06924-f003:**
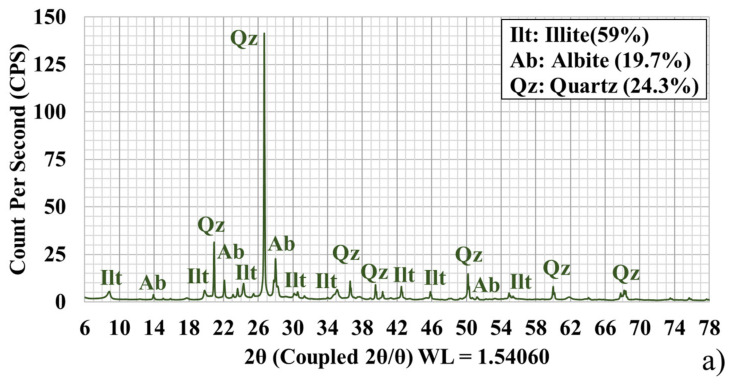
Whole-rock XRD pattern for (**a**) S08 and (**b**) S06. Low-angle XRD pattern for (**c**) S08 and (**d**) S06; their semi-quantitative abundance percentages computed with the DIFFRAC.EVA software are shown in brackets.

## Data Availability

The study did not report any data.

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
