# Peer review of "Mineral Interpretation Discrepancies Identified between Infrared Reflectance Spectra and X-ray Diffractograms"

_sensors, 2021, doi:10.3390/s21206924_

Round 1
Reviewer 1 Report
Dear authors,
Your manuscript proves that you had done a lot of work that is well presented. The novelty of your manuscript is obvious and I have some minor comments before its publication:
- Line 40 is confusing for the reader. Please rewrite.
- Line 55, please check if 6 hours is correct or you lost important amount of clay minerals as less time may be needed (3 hours).
- Line 115: Please replace Cu Kα by Cu-Kα and [AÖ¯] by Å
- The authors may had lost important peaks by their choice of scanning step from 6Ëš to 80Ëš (2θ). The scanning step must start from 2Ëš in all cases, and in fractioned samples must finish at 30Ëš.
- 3. I propose the replacement of the definition sheet silicates by clay minerals.
- Lines 153-155. This statement is not convincing. Most of montmorillonite samples have a characteristic peak at 6-7Ëš 2θ (as the authors support and they are correct), but there are some types of montmorillonite that lack this peak and present peaks close to 8Ëš 2θ (e.g. Montmorillonite 21 Å with first peak at 4 and second at 8.4Ëš 2θ). A heat treatment and an ethylene glycol treatment will provide more information for the identification of clay minerals. Due to the above, authors must avoid sentences as: “However, montmorillonite could not be confirmed in this study.”, as more experiments are needed for this conclusion.
- Line 195: The drying temperature was correctively too low to affect the crystallinity of the clay minerals.
Author Response
Dear Reviewer 1, please find the attached rebuttal letter.
Kind regards,
Fardad

Reviewer 2 Report
his paper presents the results of investigation the mineral composition of rocks containing montmorillonite and illite by X-ray diffraction (XRD) and infrared reflection spectroscopy. The main conclusions of the authors are based on their own experimental analytical data obtained using modern equipment.
The data interpretations seem to have been done well, also the data obtained may be valuable for understanding the reasons for the discrepancy between the results of both methods. So, the paper adds valuable new information concerning our knowledge on the clay minerals determination and benefits mineralogists, and geologists who study hydrothermal alteration zones defined by the presence or absence of montmorillonite and illite.I thus can recommend the paper for publication in Sencors but with minor revisions.
Comment: It is necessary to improve the quality of Figures.
Author Response
Dear Reviewer 2, please find the attached rebuttal letter.
Kind regards,
Fardad

Reviewer 3 Report
General thoughts
Line – 16-18 The conclusion is too obvious, please give a deeper meaning to the presented article. The main goal and the results obtained
Lines 22 – 38 Introduction - too general, extend with the current state of knowledge and research in the field of the compared research methods with a detailed specification of the proposed methods for the analysis of mixed-packet clay minerals.
Methodology
In X-ray analysis, there are 3 basic groups of error sources distinguished:
- related to the imperfection of the apparatus,
- related to the processes of excitation of the characteristic, X-ray radiation of a given element by the radiation of other elements in samples of complex composition,
- related to the methodology of the emitter preparing, i.e. the sample.
In the presented work, sample preparation seems to be the weakest element, especially in the context of mixed-packet clay minerals.
- Please, provide detailed information about the petrographic nature of the tested samples
- Based on what criteria, 2 of 19 samples were selected
- When identifying mixed-packet clay minerals, it is not enough to separate the clay fraction, it is also important to carry out laboratory determinations in the direction of glyocolling or roasting. Complete the methodology if there were such tests performed, if not, supplement in the future with XRD analyses on such samples.
Summary
When identifying mixed-packet minerals, only one identification method cannot be used. The work shows discrepancies between the proposed XRD and SWIR methods. Which is nothing new from a research point of view. The authors accurately analyse the obtained discrepancies. However, it should be noted that the lack of montmorillonite on XRD patterns does not mean that it is not there. Maybe it is still worth rethinking / refining the preparation of samples for XRD ...
Other remarks marked in the text.

Author Response
Dear Reviewer 3, please find the attached rebuttal letter.
Kind regards,
Fardad
